# Underlying Causes of Myocardial Infarction with Nonobstructive Coronary Arteries: Optical Coherence Tomography and Cardiac Magnetic Resonance Imaging Pilot Study

**DOI:** 10.3390/jcm11247495

**Published:** 2022-12-17

**Authors:** Joanna Fluder-Wlodarczyk, Marek Milewski, Magda Roleder-Dylewska, Maciej Haberka, Andrzej Ochala, Wojciech Wojakowski, Pawel Gasior

**Affiliations:** 1Division of Cardiology and Structural Heart Diseases, Medical University of Silesia in Katowice, 40-055 Katowice, Poland; 2Department of Cardiology, School of Health Sciences, Medical University of Silesia in Katowice, 40-055 Katowice, Poland

**Keywords:** MINOCA, STEMI, NSTEMI, CMR, OCT

## Abstract

Background: Scientific statements recommend multimodality imaging in myocardial infarction with non-obstructive coronary arteries (MINOCA) to define the underlying cause. Aim: We evaluated the diagnostic yield of intravascular optical coherence tomography (OCT) and cardiac magnetic resonance (CMR) in the MINOCA setting. Methods: In this prospective, single center, observational pilot study, we enrolled patients with MINOCA without previous coronary interventions. All patients underwent three vessel OCT, followed by CMR. Imaging results were combined to determine the mechanism of MINOCA, when possible. Results: We enrolled 10 patients in this pilot study. Women constituted 50% of the analyzed population. The mean age of patients was 52 years. ST-segment elevation was found in 30% of patients. A possible culprit lesion was identified by OCT in 70% of participants, most commonly plaque rupture or erosion. An ischemic pattern of CMR abnormalities was identified in 70% of participants. Myocarditis and Tako-Tsubo were identified in 30%. A cause of MINOCA was identified in all patients using multimodality imaging, while using OCT alone identification occurred in only 70% of patients. Conclusion: In this pilot study, multimodality imaging with OCT and CMR identified potential mechanisms in all patients with a diagnosis of MINOCA, and it has the potential to guide medical therapy for secondary prevention.

## 1. Introduction

Myocardial infarction with nonobstructive coronary arteries (MINOCA) is characterized as acute myocardial infarction (AMI) without evidence of obstructive stenosis in coronary arteries in major coronary arteries (stenosis severity < 50%) [1]. MINOCA is found in approximately 3% to 15% of patients with AMI and disproportionately affects women [2,3,4]. Although in clinical practice the underlying mechanism of MINOCA is frequently undetermined, its prognosis remains serious, with a 1-year mortality ranging from 5–10%, reaching up to 16% at 5-year follow-up [2,5]. Even though, elevated troponin levels and chest discomfort are not specific solely for AMI, MINOCA is caused by a vastly heterogeneous group of myocardial or vascular disorders, thus it should always be considered merely as a working diagnosis, which requires further investigation. To confirm MINOCA other underlying causes of elevated troponin levels must be ruled out. Therefore, finding prognostic markers and determining the specific pathophysiological mechanism is crucial to provide appropriate treatment strategies in patients with MINOCA diagnosis. Major pathophysiological mechanisms of MINOCA include plaque rupture or erosion, coronary thromboembolism, coronary artery spasm, spontaneous coronary artery dissection, Tako-Tsubo cardiomyopathy and myocarditis. Importantly, plaque disruption may be missed in coronary angiogram and occur in areas that initially appear normal [6]. Optical coherence tomography (OCT) due to its high resolution is able to precisely evaluate the morphological characteristics of atherosclerotic plaque [7]. However, cardiac magnetic resonance (CMR) allows for detection of myocardial fibrosis with a high diagnostic accuracy and is considered the gold standard for in vivo myocardial tissue characterization [5].

The purpose of this pilot study was to evaluate in a prospective cohort of patients of MINOCA the diagnostic yield of combined OCT and CMR imaging.

## 2. Materials and Methods

In this prospective, single-center study, consecutive patients aged 18 years and older who presented with suspected AMI and had no obstructive (>50% stenosis) coronary lesions on angiography and no specific alternative diagnosis for the clinical presentation were enrolled. All patients underwent OCT imaging in all major coronary arteries during the initial procedure or up to 24 h after the initial procedure. Subsequently, all patients underwent CMR during hospital stay. Major exclusion criteria were renal failure in stage ≥3, allergy to contrast media, estimated survival of <2 years, active systemic inflammatory process and pregnancy. The study was approved by Bioethical Committee and conformed to the Declaration of Helsinki. All patients signed informed consent. This study is a pilot phase of a large ongoing registry on MINOCA.

### 2.1. Coronary Angiography and Optical Coherence Tomography

Coronary angiography was performed using standard techniques through the transradial or transfemoral approach. Angiographic views were acquired in optimal projection angels using 6-Fr diagnostic catheters with manual contrast injections. Patients were excluded from participation if the angiogram showed any coronary stenosis ≥50% or excessive tortuosity and/or abnormal origin of the coronary artery that, in the opinion of the operator, increased the risk of OCT imaging. Subsequently, OCT of all major coronary arteries was performed during the initial coronary angiography or within 24 h using the iLumien OPTIS Medical system (Abbott Vascular, Santa Clara, CA, USA). OCT was performed using 6-Fr guiding catheters with manual contrast injections. Following imaging, operator performed interpretation of the OCT imaging in order to choose the most optimal treatment strategy. Subsequently OCT images were stored digitally and analyzed by 3 experienced and blinded investigators (J.F.-W., M.M., and M.R.-D.). The evaluation of the OCT images was performed according to consensus document on the acquisition, measurement and reporting of OCT studies, reported by Tearney et al. [7]. Plaque rupture (PR) was defined as the discontinuity of the fibrous cap with the evidence of presence of cavity inside the plaque [7,8]. Plaque erosion (PE) was defined as thrombus presence overlying an irregular surface without evidence of fibrous cap disruption [9]. Eruptive calcific nodule was identified in case of evident fibrous cap disruption and/or thrombus presence on the surface of calcified plaque protruding into the lumen [9]. Intraluminal thrombus was defined as irregular mass attached to the arterial wall or floating inside the lumen [7,8]. 

### 2.2. Cardiac Magnetic Resonance

The CMR images were acquired on 1.5T system Optima MR450w (GE Healthcare). All cardiac CMR imaging were electrocardiographically gated and based on the protocols developed according to guidelines and routinely used in clinical practice. 

CMR study protocol included: (1) functional sequences using cine imagining; (2) edema imaging with T2-weighted short-tau inversion recovery; and (3) viability imaging utilizing late gadolinium enhancement (LGE) imagining. At the time of interpretation, physicians responsible for CMR interpretation were not provided with results of the other imaging test. Final diagnosis was established based on CMR findings: (1) AMI: presence of subendocardial or transmural abnormalities located in the distribution of a single coronary artery, or (2) myopericarditis presence of subepicardial and/or midwall abnormalities and lack of subendocardial LGE [10]. 

### 2.3. Statistical Analysis 

Statistical analysis was performed using Statistica (Statistica v. 13, Tibco Software Inc., Palo Alto, CA, USA). Continuous variables were expressed as mean ± SD with the median and interquartile range used for variables with non-normal distributions. Categorical variables were described with percentages and counts.

## 3. Results

### 3.1. Study Population

A total of 10 patients underwent OCT imaging of the major epicardial arteries and CMR during hospitalization. The mean age of patients was 52 ± 6 years and woman constituted half of the patients. Typical angina on admission was present in 60% of patients. At admission, 40% of patients had regional wall motion abnormalities on echocardiography and the mean left ventricle ejection fraction (LVEF) was 51 ± 13%. None of the patients had a severe valvular heart disease. Mean troponin level at admission was 0.24 ± 0.18 ng/mL, while maximal level was (0.35 ± 0.22 ng/mL). Mean low-density lipoprotein level at admission was 97.60 ± 30.03 mg/dL. In total, 70% of patients presented as NSTEMI. The baseline characteristics are summarized in Table 1. In Appendix A, baseline characteristics are summarized for each patient.Coronary arteries were normal in 40% of patients and the remaining patients presented with mild to moderate (up to 50%) coronary lesions. 

### 3.2. OCT and CMR Findings

The number of OCT pullback acquisitions per patient was 5.2 ± 0.8. OCT provided clear substrate for AMI in 70% of patients and remaining patients had negative result. Plaque rupture, plaque erosion, and spontaneous coronary artery dissection (SCAD) were found in 20%, 30% and 20% patients, respectively. Thrombus was present in all patients with plaque rupture and in one patient with plaque erosion. No eruptive calcific nodules were found in any patient. Average contrast volume used per patient was 102 ± 8 mL. There were no complications during OCT examination. In the CMR imaging, AMI was evident in 70% of patients. In addition, substrate for AMI in OCT was found in all coronary arteries supplying the infarct-related territory as confirmed by CMR. There was a complete concordance between new ischemic lesions in CMR and presence of positive OCT findings. In 90% of patients, T2 hyperintensity was found. The mean number of segments with LGE was 1.7 ± 0.7. Myocarditis was diagnosed using CMR in two patients and Tako-Tsubo in one subject. One patient had microvascular obstruction. Pericardial effusion was found in two patients. OCT and CMR findings are summarized in Table 2. In Appendix A, OCT and CMR findings are summarized for each patient. A representative picture of multimodality imaging is presented in Figure 1.

### 3.3. Treatment at Discharge

All patients received Aspirin at discharge and P2Y12 inhibitors were prescribed in 80% of patients. No patient received oral anticoagulants, but one patient received low molecular weight heparin at discharge. Long-acting nitrates were prescribed in one case. Eight patients received angiotensin-converting enzyme inhibitors and all included subjects were prescribed with beta-blockers at discharge. All patients received statins. The summary of medications at discharge is presented in Table 3.

## 4. Discussion

The results of our analysis demonstrates that OCT combined with CMR imaging ensures identification of MINOCA substrate in all patients presenting with AMI without obstructive lesions in coronary arteries. Furthermore, OCT findings well corresponded with CMR confirmed myocardial injury. Our results emphasize the benefit of using additional testing for more detailed evaluation MINOCA patients, which may potentially lead to better tailored treatment based on the actual underlying pathophysiological mechanism.

It must be stressed that MINOCA is not a benign condition; it has a 3-year mortality reaching up to 16% [2]. The European Association of Percutaneous Coronary Interventions recently published an expert consensus document, which strongly recommends adoption of intracoronary imaging to complete diagnosis in cases where uncertainty exists based solely on angiography in the AMI setting [11]. The most frequent event leading to AMI is coronary thrombosis, and while filling defects and haziness observed in angiography may suggest present of plaque disruption, OCT is able to provide definite diagnosis due to its superior resolution. Furthermore, a large thrombus usually causes significant stenosis or even occlusion of coronary artery, whereas smaller thrombi are usually associated with insignificant narrowing (not visible in angiography) or distal segment embolization [12,13]. Information regarding the exact pathological mechanism responsible for MINOCA presentation often cannot be obtained based on angiographical evaluation alone. In the previously published studies PR rates varied between 24–34% among patients with working diagnosis of MINOCA, which is similar to the presented study (20%) [14,15]. However, existing evidence suggest that PR is not the only cause that may lead to acute vessel thrombosis. Recent studies demonstrated that in up to 30% of patients with ST-segment elevation myocardial infarction there was no evidence of PR, but rather presented with PE [9,16,17,18]. In our study, the rate of PE (20%) was similar to previously published results ranging from 11–31% in patients with suspected MINOCA [14,15]. Furthermore, in our study, the prevalence of coronary artery dissection was 20%, which is higher than in a recently published OCT and CMR study (5%) [19]. Spontaneous coronary artery dissection may be missed or even misdiagnosed as vasospasm on angiogram due to low resolution of this modality, even though it might be a life-threating condition [20]. The main advantage of the presented analysis is that all patients underwent three vessel OCT together with CMR imaging.

The clinical use of CMR is increasingly growing due to its ability to confirm MI and accurately categorize patients according to the contemporary definition of MINOCA [21]. Furthermore, CMR is a noninvasive technique and can precisely differentiate MI from other conditions including myocarditis and various cardiomyopathies, which significantly enhances patient diagnosis, personalized treatment pathways, and assessment of prognosis. In our study, CMR showed high rate of ischemic LGE (70%), which is similar to the results previously reported in the MINOCA population (77.5%) [15]. In addition, in this prospective study, OCT provided clear substrate for AMI in all coronaries supplying the infarct-related region as confirmed by CMR. Our results demonstrated that high-risk lesions identified by OCT might indicate atherosclerosis as an underlying factor of AMI confirmed by CMR in the MINOCA population. 

There is a paucity of data on recommended secondary prevention therapies for patients with MINOCA [1,3,22]. Guidelines stress the need for individualized treatment based on the underlying mechanism of MINOCA [21]. A recent, large, retrospective study demonstrated that use of statins, ACEI/ARB and beta blockers was associated with a reduction in the adverse events rate [23]. In our study, all patient received b-blockers and statins and in a majority of cases, ACEI/ARB were prescribed upon discharge. Furthermore, it needs to be stressed that randomized trials investigating secondary prevention in MINOCA setting are of utmost clinical importance, especially when considering the uncertainty regarding underlying pathophysiological mechanisms and the large number of patients with adverse long-term prognosis.

The major finding of this study is fact that combination of CMR and OCT provides a clear substrate and/or diagnosis in all patients with MINOCA. Therefore, the widespread adoption of the multimodality imaging can facilitate diagnostic process and allows the physicians to choose individualized treatment strategies, which might be challenging in daily practice in some regions due to limited CMR and OCT availability. In addition, providing patient with a definite diagnosis could also improve compliance with therapeutic strategies. However, whether combination of CMR and OCT-guided therapy may improve outcomes in patients presenting with a working diagnosis of MINOCA has to be further investigated in dedicated large randomized studies.

### Study Limitations

This study has some important limitations that need to be stressed. First, this prospective, observational, single center pilot study included a small number of patients. Second, patient selection bias cannot be excluded as we were unable to enroll all eligible patients due to logistical problems. Third, OCT was not systematically performed during the initial coronary angiography. Finally, invasive physiological evaluation was not included in our research protocol.

## 5. Conclusions

In this pilot study, OCT coupled with CMR provided a clear substrate and/or diagnosis in all patients presenting with MINOCA, emphasizing the need for multimodality imaging for tailored treatment in this population.

## Figures and Tables

**Figure 1 jcm-11-07495-f001:**
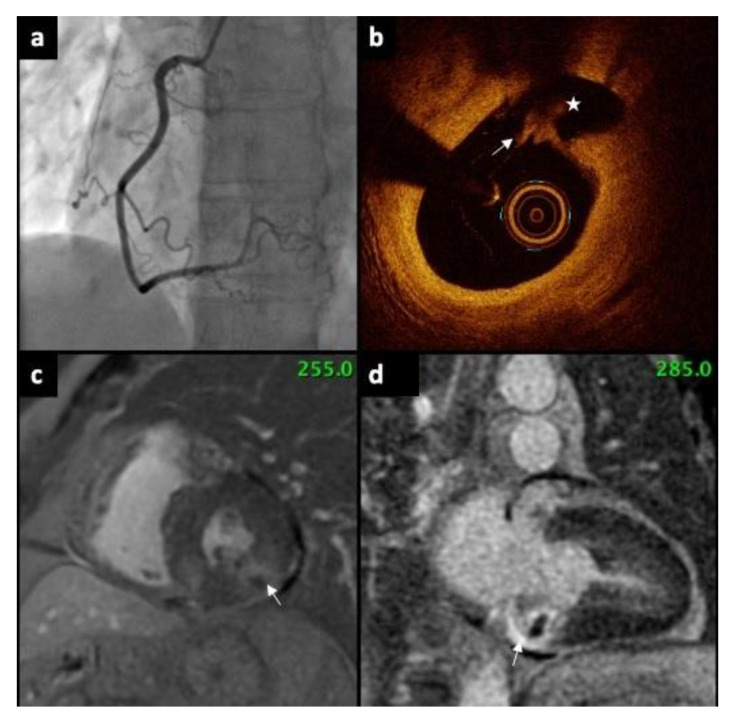
(**a**) Coronary angiography showing normal right coronary artery, without any stenosis. (**b**) OCT revealed ruptured plaque (arrow). Rupture cavity (asterisk) is formed behind fibrous cap disruption (**c**) CMR (short axis) presents subendocardial myocardial LGE (arrow). (**d**) CMR (LV two-chamber, long axis) demonstrate subendocardial scar in the basal segment of inferior wall (arrow). OCT—Optical Coherence Tomography, CMR—Cardiac magnetic resonance, LGE—late gadolinium enhancement, LV—left ventricle.

**Table 1 jcm-11-07495-t001:** Baseline characteristics.

	*n* = 10
Age (years [Q1–Q3])	53 (49–55)
Female, *n* (%)	5 (50)
NSTEMI, *n* (%)	7 (70)
Typical angina (%)	6 (60)
SBP (mmHg [Q1–Q3])	130 (110–150)
DBP (mmHg [Q1–Q3])	80 (75–100)
HR (1/min [Q1–Q3])	75 (70–90)
LVEF (% [Q1–Q3])	55 (40–60)
RWMA, *n* (%)	4 (40)
Severe valvular disease, *n* (%)	0
Pericardial effusion, *n* (%)	1 (10)
NYHA I, *n* (%)	6 (60)
NYHA II, *n* (%)	4 (40)
NYHA III, *n* (%)	0
NYHA IV, *n* (%)	0
Hypertension, *n* (%)	8 (80)
Hypercholesterolemia, *n* (%)	2 (20)
Smoking, *n* (%)	7 (70)
Diabetes, *n* (%)	1 (10)
Obesity, *n* (%)	1 (10)
Previous CAD, *n* (%)	0
Previous stroke, *n* (%)	0
Previous kidney disease, *n* (%)	0
Previous lung disease, *n* (%)	0
Family history, *n* (%)	3 (30)
Sinus rhythm in ECG, *n* (%)	10 (100)
Elevated C reactive protein > 5 mg/L, *n* (%)	3 (30)
Troponin at admission (ng/mL)	0.24 ± 0.18
Troponin maximal level (ng/mL)	0.35 ± 0.22
Creatinine at admission (mg/dL)	0.78 ± 0.20
Glucose at admission (mg/dL)	118.70 ± 40.84
Total cholesterol at admission (mg/dL)	172.10 ± 30
LDL at admission (mg/dL)	97.60 ± 30.03
Triglycerides at admission (mg/dL)	118.90 ± 30.87

NSTEMI = non-ST segment elevation myocardial infarction. SBP = systolic blood pressure. DBP = diastolic blood pressure. HR = heart rate. LVEF = left ventricle ejection fraction. RWMA = regional wall motion abnormalities, CAD = coronary artery disease, LDL = low-density lipoprotein.

**Table 2 jcm-11-07495-t002:** OCT and CMR findings.

OCT	
Plaque rupture, *n* (%)	2 (20)
Plaque erosion, *n* (%)	2 (20)
Eruptive calcific nodule, *n* (%)	0 (0)
Spontaneous coronary artery dissection, *n* (%)	2 (20)
Presence of thrombus, *n* (%)	3 (30)
Negative OCT, *n* (%)	3 (30)
**CMR**	
T2 hyperintensity	
Present, *n* (%)	9 (90)
Absent, *n* (%)	1 (10)
Myocardial hemorrhage, *n* (%)	0 (0)
MVO, *n* (%)	1 (10)
Abnormal delayed enhancement	
Subendocardial, *n* (%)	5 (50)
Transmural, *n* (%)	7 (70)
Subendocardial + transmural, *n* (%)	4 (40)
LV dimensions and function	
LVEDV (mL/m^2^)	125 ± 25
LVESV (mL/m^2^)	50 ± 16
LVEF (%)	60 ± 8
Pericardial effusion, *n* (%)	2 (20)
Definite diagnosis	
MI, *n* (%)	7 (70)
Myocarditis, *n* (%)	2 (20)
Negative CMR, *n* (%)	1 (10)

OCT—optical coherence tomography, CMR—cardiac magnetic resonance, MVO—microvascular obstruction, LV—left ventricle, MI—myocardial infarction.

**Table 3 jcm-11-07495-t003:** Medications at discharge.

Aspirin, *n* (%)	10 (100)
P2Y12 inhibitors, *n* (%)	8 (80)
ACEI/ARB, *n* (%)	8 (80)
Beta-blockers, *n* (%)	10 (100)
MRA, *n* (%)	2 (20)
Statins, *n* (%)	10 (100)
Nitrates, *n* (%)	1 (10)
LMWH, *n* (%)	1 (10)
Oral anticoagulants, *n* (%)	0 (0)

ACEI = angiotensin converting enzyme inhibitors, ARB—angiotensin receptor blockers. LMWH = low molecular weight heparin. MRA = mineralocorticoid receptor antagonists.

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
