# Peer review of "Underlying Causes of Myocardial Infarction with Nonobstructive Coronary Arteries: Optical Coherence Tomography and Cardiac Magnetic Resonance Imaging Pilot Study"

_jcm, 2022, doi:10.3390/jcm11247495_

Round 1

Reviewer 1 Report

In this paper Fluder-Wlodarczyk J, et al. describe a pilot study, which demonstrated that multi-modality imaging with OCT and CMR identified potential mechanisms in all patients with a diagnosis of MINOCA and has the potential to guide medical therapy for secondary prevention.

In general, I liked the idea proffered by the authors. The bi-modality use by the authors was found to be effective in delineating the pathogenesis of the acute event, or so-called MINOCA. This is not a novel concept as the authors themselves report other similar studies in the discussion paragraph. Yet, the concept is an appealing one as long as you have both modalities of OCT and CMR at your disposal.

My main objection resides in the definition of MINOCA applied by the authors. In the first sentence of the Materials and Methods section they refer to the patients included as “AMI patients but with no obstructive coronary lesions on angiography….”. The authors write: “Myocardial infarction with nonobstructive coronary arteries (MINOCA) is characterized as acute myocardial infarction (AMI) without evidence of obstructive stenosis in coronary arteries in major coronary arteries (stenosis severity <50%).”

This is a misinterpretation of the clinical presentation. The fact that troponin is elevated is not sufficient to define these patients as AMI patients with an underlying pathophysiologic mechanism to be investigated and revealed. First and foremost, patients with myocarditis or mere cardiac injury due to and infectious or inflammatory disease should be eliminated. To this end, the authors should present their CRP values.   

Presentation of Table 1, better to present incidence as N (%). It is a clearer representation and shows clearly that the numbers of patients evaluate is low, that is alright since we’re dealing with a small pilot study and this fact is also mentioned in the limitation section.

As discussed above we recommend that Table 1 should include CRP concentration. A priori and certainly post-factum the clinical and biochemical presentation fits, at least theoretically, the diagnosis of peri-myocarditis.

We can fully agree with the authors of the paper that: “The major finding of this study is fact that combination of CMR and OCT provides a clear substrate and/or diagnosis in all patients with MINOCA.”, though we should realize that OCT is not universally available.

Typos and Style

“Average contrast use volume per patient was 102±8ml.” Maybe the authors meant to say: “contrast volume use was…”?

Figure legend: “A. Coronary angiography showing normal right coronary artery, without any stenosis. B. 139 OCT revealed raptured plaque (arrow). Rapture cavity (asterisk) is formed behind fibrous cap disruption…. Obviously, minor typo: should be “rupture”.

European Association of Percutaneous Coronary Interventions recently published an expert consensus document which strongly recommends adoption intracoronary imaging to complete diagnosis in cases where uncertainty exists based solely on angiography in the AMI setting. Again minor style issue: “…..recommends adoption of intracoronary imaging to complete establish diagnosis in cases….”

“..whereas smaller thrombi is usually associated with insignificant narrowing (not visible in angiography) or distal segment embolization…” thrombi is are associated with….”

The sentence “Our results demonstrated that OCT confirmed high-risk lesions can indicate atherosclerosis as a causative factor of CMR confirmed AMI in the MINOCA population.” to me is somewhat unintelligible and should be rephrased for the benefit of the readers.  

Author Response

Response to Reviewer

In this paper Fluder-Wlodarczyk J, et al. describe a pilot study, which demonstrated that multi-modality imaging with OCT and CMR identified potential mechanisms in all patients with a diagnosis of MINOCA and has the potential to guide medical therapy for secondary prevention.

In general, I liked the idea proffered by the authors. The bi-modality use by the authors was found to be effective in delineating the pathogenesis of the acute event, or so-called MINOCA. This is not a novel concept as the authors themselves report other similar studies in the discussion paragraph. Yet, the concept is an appealing one as long as you have both modalities of OCT and CMR at your disposal.

QUERY#1: My main objection resides in the definition of MINOCA applied by the authors. In the first sentence of the Materials and Methods section they refer to the patients included as “AMI patients but with no obstructive coronary lesions on angiography….”. The authors write: “Myocardial infarction with nonobstructive coronary arteries (MINOCA) is characterized as acute myocardial infarction (AMI) without evidence of obstructive stenosis in coronary arteries in major coronary arteries (stenosis severity <50%).”

This is a misinterpretation of the clinical presentation. The fact that troponin is elevated is not sufficient to define these patients as AMI patients with an underlying pathophysiologic mechanism to be investigated and revealed. First and foremost, patients with myocarditis or mere cardiac injury due to and infectious or inflammatory disease should be eliminated. To this end, the authors should present their CRP values.  

RESPONSE: We thank reviewer for this comment. We fully agree that non-ischemic causes of MINOCA must also be considered. Several disorders that result in myocardial injury (ie myocarditis, pulmonary embolism and some cardiomyopathies) may mimic ischemic MI and fulfil the universal criteria for AMI. This was stressed in the introduction section in the sentence “Even though, elevated troponin levels and chest discomfort are not specific solely for AMI, MINOCA is caused by a vastly heterogeneous group of myocardial or vascular disorders, thus it should always be considered merely as a working diagnosis which requires further investigation”. In fact, in our population multimodality imaging with OCT and CMR allowed to diagnose myocarditis in 20% of included patients.  Furthermore, we’ve modified the material and methods section mentioned by the reviewer “In this prospective, single-centre study, consecutive patients aged 18 years and older who presented with suspected AMI diagnosis and had no obstructive (>50% stenosis) coronary lesions on angiography and no specific alternative diagnosis for the clinical presentation were enrolled”. Finally, we added CRP values to the Table 1.

QUERY#2: Presentation of Table 1, better to present incidence as N (%). It is a clearer representation and shows clearly that the numbers of patients evaluate is low, that is alright since we’re dealing with a small pilot study and this fact is also mentioned in the limitation section.

RESPONSE: We modified Table 1 according to reviewers suggestion.

QUERY#3: As discussed above we recommend that Table 1 should include CRP concentration. A priori and certainly post-factum the clinical and biochemical presentation fits, at least theoretically, the diagnosis of peri-myocarditis.

RESPONSE: As requested by the reviewer we added CRP levels.

QUERY#4: We can fully agree with the authors of the paper that: “The major finding of this study is fact that combination of CMR and OCT provides a clear substrate and/or diagnosis in all patients with MINOCA.”, though we should realize that OCT is not universally available.

RESPONSE: We thank reviewer for this comment. We modified the following sentence in order to highlight limited OCT and CMR availability in some regions Therefore, the widespread adoption of the multimodality imaging can facilitate diagnostic process and allows the physicians to choose individualized treatment strategies, which might be challenging in daily practice in some regions due to limited CMR and OCT availability.

QUERY#5: Typos and Style

“Average contrast use volume per patient was 102±8ml.” Maybe the authors meant to say: “contrast volume use was…”?

Figure legend: “A. Coronary angiography showing normal right coronary artery, without any stenosis. B. 139 OCT revealed raptured plaque (arrow). Rapture cavity (asterisk) is formed behind fibrous cap disruption…. Obviously, minor typo: should be “rupture”.

European Association of Percutaneous Coronary Interventions recently published an expert consensus document which strongly recommends adoption intracoronary imaging to complete diagnosis in cases where uncertainty exists based solely on angiography in the AMI setting. Again minor style issue: “…..recommends adoption of intracoronary imaging to complete establish diagnosis in cases….”

“..whereas smaller thrombi is usually associated with insignificant narrowing (not visible in angiography) or distal segment embolization…” thrombi is are associated with….”

RESPONSE: We corrected the mentioned typos and stylistic errors.

QUERY#6: The sentence “Our results demonstrated that OCT confirmed high-risk lesions can indicate atherosclerosis as a causative factor of CMR confirmed AMI in the MINOCA population.” to me is somewhat unintelligible and should be rephrased for the benefit of the readers.

RESPONSE: We thank reviewer for this comment. We rephrased the mentioned sentence Our results demonstrated that high-risk lesions identified by OCT might indicate atherosclerosis as an underlying factor of AMI confirmed by CMR in the MINOCA population.

Reviewer 2 Report

I read with great interest the article by Fluder-Wlodarczyk J, et al: " Underlying Causes of Myocardial Infarction with Nonobstructive Coronary Arteries: Optical Coherence Tomography and Cardiac Magnetic Resonance Imaging Pilot Study" This study addresses the problem of MINOCA. The pilot study investigated whether multimodality imaging can provide additional information in patients with MINOCA. The paper nicely illustrates the importance of multimodal imaging in patients with MINOCA. The main conclusion of the study is that multimodality imaging is necessary for tailored treatment in this patient group. The data presented are interesting and the authors address all possible limitations. In particular, they provide additional information to the literature data on multimodality imaging in this patient group.

The study mainly addressed whether optical coherence tomography (OCT) in combination with cardiac magnetic resonance (CMR) imaging can better identify the cause of acute coronary syndrome in patients with normal coronary angiography.

The issue is important in this area because most patients with MINOCA today receive only angiography and possibly CMR later. This study has shown that OCT should be performed during diagnostic angiography if the coronary angiography result is normal. In addition, OCT could guide therapy (eg, in spontaneous coronary artery dissection). Up to now, the OCT had the recomendation that "intravascular ultrasound or OCT may be useful." This finding suggests that the recommendation could be changed to "should be considered." To date, there are no randomized control trials on this issue.

Compared with other published material, this study emphasizes the usefulness of OCT in combination with CMR in these patients. Conclusions are consistent with the evidence and arguments presented. But this was a pilot study with a very small number of participants. A study with a larger number of participants is needed. 

Author Response

Response to Reviewer

QUERY#1:

I read with great interest the article by Fluder-Wlodarczyk J, et al: " Underlying Causes of Myocardial Infarction with Nonobstructive Coronary Arteries: Optical Coherence Tomography and Cardiac Magnetic Resonance Imaging Pilot Study" This study addresses the problem of MINOCA. The pilot study investigated whether multimodality imaging can provide additional information in patients with MINOCA. The paper nicely illustrates the importance of multimodal imaging in patients with MINOCA. The main conclusion of the study is that multimodality imaging is necessary for tailored treatment in this patient group. The data presented are interesting and the authors address all possible limitations. In particular, they provide additional information to the literature data on multimodality imaging in this patient group.

The study mainly addressed whether optical coherence tomography (OCT) in combination with cardiac magnetic resonance (CMR) imaging can better identify the cause of acute coronary syndrome in patients with normal coronary angiography.

The issue is important in this area because most patients with MINOCA today receive only angiography and possibly CMR later. This study has shown that OCT should be performed during diagnostic angiography if the coronary angiography result is normal. In addition, OCT could guide therapy (eg, in spontaneous coronary artery dissection). Up to now, the OCT had the recomendation that "intravascular ultrasound or OCT may be useful." This finding suggests that the recommendation could be changed to "should be considered." To date, there are no randomized control trials on this issue.

Compared with other published material, this study emphasizes the usefulness of OCT in combination with CMR in these patients. Conclusions are consistent with the evidence and arguments presented. But this was a pilot study with a very small number of participants. A study with a larger number of participants is needed.

RESPONSE:

We fully agree with the reviewer. We modified the last sentence in the discussion section in order to emphasize the need for a large randomized trial in this subject “However, whether combination of CMR and OCT-guided therapy may improve outcomes in patients presenting with working diagnosis of MINOCA has to be further investigated in dedicated large randomized studies.”

Reviewer 3 Report

Fluder-Wlodarczyk and coll. Performed an interesting study exploring the Underlying Causes of Myocardial Infarction with Nonobstructive Coronary Arteries. The topic is interesting and the manuscript well written.  I have some minor comments:

-          The methods must be improved adding how the coronary angiography was performed

-          I suggest adding, as supplementary files, the characteristics reported in Tables 1 and 2 for each single patients.

-          The authors stated that is a prospective study, but there are no data for the follow-up.

-          Did some patients perform FFR or IFR? This aspect must be also discussed

-          A revision for English grammar and typo errors must be performed.

Author Response

Response to Reviewer

Fluder-Wlodarczyk and coll. Performed an interesting study exploring the Underlying Causes of Myocardial Infarction with Nonobstructive Coronary Arteries. The topic is interesting and the manuscript well written.  I have some minor comments:

QUERY#1:The methods must be improved adding how the coronary angiography was performed

RESPONSE: We thank reviewer for the suggestion, we elaborated on the mentioned subject in the materials and methods section Coronary angiography was performed using standard techniques through the transradial or transfemoral approach. Angiographic views were acquired in optimal projection angels using 6-Fr diagnostic catheters with manual contrast injections. Patients were excluded from participation if the angiogram showed any coronary stenosis ≥50% or excessive tortuosity and/or abnormal origin of the coronary artery that, in the opinion of the operator, increased the risk of OCT.

QUERY#2: I suggest adding, as supplementary files, the characteristics reported in Tables 1 and 2 for each single patients.

RESPONSE: Supplementary tables were added according to reviewers suggestions.

QUERY#3: The authors stated that is a prospective study, but there are no data for the follow-up.

RESPONSE: We thank reviewer for this comment. However, the purpose of our study was to evaluate in a prospective cohort of patients of MINOCA the diagnostic yield of combined OCT and CMR imaging. Evaluation of patient outcomes wan not a purpose of this pilot study and was not reported in this communication. Patient outcomes results will be published in the future as a summary of a large ongoing registry of MINOCA patients.

However, at this point we collected follow-up via phone call. The median follow up was 12 (9-14) months. One patient was lost to follow-up. Also, one patient died from unknown causes one month following initial hospitalization. No other adverse cardiac events or unplanned hospitalizations were reported during observation period.

QUERY#4: Did some patients perform FFR or IFR? This aspect must be also discussed

RESPONSE: FFR and iFR were not performed. This was stated in the limitation section “Finally, invasive physiological evaluation was not included in our research protocol.”

QUERY#5: A revision for English grammar and typo errors must be performed.

RESPONSE: We thank reviewer for this comment. We revised manuscript and corrected grammar and typos.

Round 2

Reviewer 3 Report

No furhter comments